# Management of Stress Urinary Incontinence by Obstetricians and Gynecologists in Jordan: A Nationwide Survey Study

**DOI:** 10.3390/healthcare12151489

**Published:** 2024-07-27

**Authors:** Ayman Qatawneh, Fatemah N. Lari, Wedad A. Sawas, Fatemah A. Alsabree, Mariam Kh. Alowaisheer, Marah A. Aldarawsheh, Renad A. Alshareef

**Affiliations:** 1Department of Obstetrics and Gynecology, The University of Jordan, Amman 11942, Jordan; 2Department of Medicine, The University of Jordan, Amman 11942, Jordan; fatimaa_lari@hotmail.com (F.N.L.); wedadsawas@gmail.com (W.A.S.); fatemah.alsabree@hotmail.com (F.A.A.); mkalowaisheer98@gmail.com (M.K.A.); m_aldarawsheh@yahoo.com (M.A.A.); renad.abdullah.alshareef@gmail.com (R.A.A.)

**Keywords:** stress urinary incontinence, urogynecology, survey study, Jordan

## Abstract

Background: Stress urinary incontinence (SUI) is a common condition that can significantly impact a patient’s quality of life. Although multiple diagnostic and treatment options exist, significant variability in SUI management exists between countries. Since women’s SUI prevalence in Jordan is high, and Jordan is a lower-middle-income country, this study aimed to investigate how obstetricians and gynecologists (OBGYNs) across Jordan manage and treat women with SUI. Method: A Google Forms survey was prepared and sent out to Jordanian OBGYNs via WhatsApp. The results were collected and arranged in Microsoft Excel and then transferred to SPSS for statistical analysis. Results: Out of the 804 Jordanian registered OBGYNs, 497 could be reached, 240 conduct gynecological surgeries, and 94 completed the survey, providing a response rate of 39.2%. Most of the respondents were females between 41 and 55 years old. More than 70% of the OBGYNs worked in the private sector, and 88.3% operated in the capital of Jordan. Most of the respondents favored lifestyle and behavior therapy (43.6%) or pelvic floor physiotherapy (40.4%) as the first-line management for SUI. The transobturator mid-urethral sling (MUS) was the most common initial surgical treatment option. The physicians preferred two-staged procedures for the repair of pelvic organ prolapse alongside concomitant SUI. In the case of recurrent SUI following surgery, 77% of the respondents chose to refer to a urologist or urogynecologist. Conclusions: The Jordanian OBGYNs preferred using lifestyle/behavioral therapy and pelvic floor muscle physiotherapy as the first-line treatment to manage SUI. Secondly, the MUS would be the most frequently preferred surgical choice. To effectively manage SUI, adequate training in urogynecology and referral resources are essential in lower-middle-income countries.

## 1. Introduction

Urinary incontinence is a common global problem affecting around 423 million people over 20 years old [1], indicating a high incidence and prevalence. Furthermore, urinary incontinence is considered to be a major challenge for the health system. Urinary incontinence has different forms, including urge urinary incontinence and stress urinary incontinence (SUI) [2,3]. Urge urinary incontinence affects men, whereas SUI dominates in women [2,3]. According to the International Urogynecological Association (IUGA)/International Continence Society (ICS), SUI is defined as the involuntary leakage of urine upon effort, physical exertion, coughing, or sneezing [4,5]. The two main mechanisms behind SUI are urethral hypermobility, which occurs due to the loss of support from the pelvic floor muscles or vaginal connective tissue, and intrinsic sphincter deficiency, which occurs due to the loss of the mucosal and muscular tone in the urethra and tends to be more severe [6,7]. SUI affects the quality of life of millions of women worldwide, affecting their physical, social, and sexual health, thereby causing anxiety, depression, and the urge to withdraw from daily activities [8]. Therefore, it is paramount that viable treatment options with high curative rates and low complication rates are offered to women to improve their quality of life. 

Conservative measures and surgical methods are recommended to patients to improve their quality of life [9,10,11]. The patient’s assessment begins with a focused history and physical exam and can include different investigations, such as post-residual urine volume, urine testing, and urodynamic testing [10]. Similarly, the management can range from conservative options such as pelvic floor exercises with or without biofeedback that support dynamic lumbopelvic stabilization and lumbar muscle resistance training to various surgical procedures such as bladder neck injections, mid-urethral slings, and colposuspension [9,10,11]. 

Many scientific groups or institutions have already developed SUI diagnosis and management recommendations. Nevertheless, regional variations in practice patterns exist, especially in lower-middle-income countries (LMICs) [12]. With the different diagnostic and treatment modalities available, coupled with the bans on polypropylene mesh in certain countries, it is crucial to understand and analyze how physicians worldwide and from LMICs treat SUI [13]. In Jordan, which is an LMIC, several studies reported high SUI prevalence among women [14,15,16], with approximately 37% and 16% reporting moderate and severe symptoms, respectively [16]. Thus, the present study aimed to investigate how obstetricians and gynecologists (OBGYNs) in Jordan manage and treat women with SUI. 

## 2. Methods

### 2.1. Study Design

This cross-sectional study was conducted from 20 December 2022 to 9 March 2023 using an online Google Forms-based survey questionnaire (see Appendix A). The investigators used the Jordanian Medical Association to obtain the listed and registered OBGYNs in Jordan. It is understandable that not all registered OBGYNs perform gynecological surgeries; however, all were reached by the investigators using the study link via WhatsApp. Participants had to meet specific eligibility criteria; the most important criterion was having seen and treated women with SUI. The study’s objectives and voluntary nature were explained on the first page of the questionnaire.

The survey, a collaborative effort with esteemed OBGYNs, comprised 19 questions divided into three main sections. The first part focused on collecting demographic and specialty data, where participants were asked to provide basic information such as age, gender, practice location, training level, and work experience. The second part delved into the management and diagnostic aspects of SUI, such as preferred diagnostic modalities preoperatively and first line of management. The final section consisted of questions about the surgical treatment option that the participant would choose in different scenarios. The full survey can be found in Appendix A.

### 2.2. IRB Approval

The research study was approved by the Institutional Review Board RB at the Jordan University hospital (11/2023/9367). On the questionnaire’s first page, it was written that “the data provided will only be used for research purposes and will be dealt with confidentially. None of the questions address your identity, which will be anonymous throughout. Proceeding in completing this questionnaire will be taken as consent to participate in this study, and you are always free to withdraw at any point”.

### 2.3. Statistical Analysis

Data were collected and transferred to a Microsoft Excel (Version 2406) spreadsheet. After coding and through double-checking, the data were exported to SPSS version 26 for statistical analysis. The validity of the questionnaire was tested using the 2-tailed Pearson correlation. Frequencies, percentages, cross-tabulations, and chi-squared analyses were conducted on continuous variables. In addition, a univariate analysis was conducted to test for association between each dependent question and fixed parameter. A *p*-value of ≤0.05 was considered significant.

## 3. Results

Out of the 804 registered Jordanian OBGYNs, 497 could be reached, and 240 performed gynecological surgeries and therefore were eligible for the study. Out of the 240 OBGYNs, 94 completed the survey, providing a response rate of 39.2%. The majority of the OBGYNs were between 41 and 55 years old, and the female to male ratio was ~1.8 (Table 1). More than 70% of the respondents worked in the private sector, 88.3% worked in the capital of Jordan, Amman, and a few, 8.5%, had a urogynecology specialty. Over 58% had no subspeciality.

Most of the physicians favored lifestyle and behavior therapy (43.6%) or pelvic floor physiotherapy (40.4%) as the first-line management for SUI (Figure 1). On the contrary, very few respondents (14.9%) would go for surgery, and only (1%) would start with medications to manage SUI. No significant association was evident between the first-line management and the physician’s age group, gender, specialty, years out of fellowship, or practice setting.

The physicians were asked about their investigations of choice before the surgical management of SUI. In this question, three processes were preferred; 26.6% (n = 25) would only perform a physical exam, and an equal amount of other respondents (26.6%) would perform stress testing, uroflow, and measure the post-residual urine volumes, and 24.5% would conduct a multichannel urodynamics test (Figure 2). To a lesser extent, 9.6% would perform a stress test and post-void residual volume, 6.4% a stress test only, and 6.4% video urodynamics. Interestingly, all six respondents who chose video urodynamics as their initial presurgical diagnostic modality worked in private practice clinics.

No significant association was established between the diagnostic modality preference and the physician’s age group, gender, specialty, years out of fellowship, physician practice setting, or location. 

When the OBGYNs were asked if they would repeat the urodynamics before a second operation, a significant association was found between repeating the urodynamics and the OBGYN’s age group (*p* < 0.01). Seventy-one percent (71%) of the 30–40-year-old age group responded to repeat urodynamics testing all the time compared to 56%, 36%, and 14% for the 41–55, >50, and <30 age groups. 

The primary surgical treatment option for the OBGYNs in Jordan was the transobturator mid-urethral sling (MUS-TO) procedure (46.8%), followed by the retropubic mid-urethral sling (MUS-RP) approach (17%), and Kelly plication (17%) was the next option in terms of frequency (Figure 3). On the other hand, bladder neck needle suspension and the Burch procedure were preferred by only 8.5% and 5.3%, respectively. Most of the physicians (79%) who had fellowship or subspecialty training favored the MUS-TO compared to 39% who did not have fellowship or OBGYN subspecialty training. These differences were significant (*p* < 0.01).

The OBGYN’s age group and gender were significantly associated with the choice regarding the primary surgical option (*p* < 0.01 and *p* < 0.01). The young OBGYNs were more prone to use the MUS-TO than older physicians. Interestingly, the percentages of using the MUS-TO were 71%, 64%, 44%, and 6% for the age groups <30, 30–40, 41–55, and >55. In addition, the male OBGYNs (62%) were more prone to use the MUS-TO compared to 38.2% of the female OBGYNs. No significant association was apparent between the primary surgical option and the specialty, years out of fellowship, or practice setting.

Next, the physicians were asked about urodynamic testing in the setting of repeat surgery. More than half (52.1%) indicated that they would always repeat the urodynamic testing, whereas 42.6% would repeat it some of the time, and 5.3% would never repeat it. Although many different treatment options were included in the questionnaire for the secondary surgical management of SUI after the failure of the primary procedure, most of the respondents (77%) stated that they would refer the patient to either a urologist or urogynacologist as a part of their management. 

In the case of SUI with concomitant pelvic organ prolapse (POP), 34% of the Jordanian OBGYNs indicated that they would perform two-staged procedures, repairing the POP first, followed by SUI surgery at a later date, if indicated. Of the practitioners that would perform the surgery at the time of the POP repair, the MUS-TO was the most popular choice (21.3%), followed by Kelly plication (14.9%), the MUS-RP (9.6%), the Burch procedure (7.4%), bladder neck needle suspension (7.4%), single incision slings (4.3%), and lastly a urethral bulking agent (1.1%). No significant association was found between the preferred surgical treatment of the SUI option and the physician’s age, gender, specialty, practice setting, fellowship, or years out of fellowship. 

Furthermore, regarding the preferred surgical treatment to prevent de novo SUI when performing a simultaneous POP surgery in the absence of symptoms and objective evidence of SUI, the majority (66%) of the physicians indicated that they would not perform surgery. Out of the latter, most of them were female OBGYNs (*p* < 0.05) and those who did not have fellowship or subspecialty training (*p* < 0.05). On the other hand, some of the respondents (9.6%) chose Kelly plication, 8.5% selected the MUS-TO (8.5%), and 5.3% opted for the Burch procedure. The rest chose the MUS-RP (4.3%), single incision slings (3.2%), urethral bulking agents (2.1%), and bladder neck needle suspension (1.1%). 

In the last two questions, the physicians were asked about the MUS material and the choice of the urethral bulking agent. Almost all the physicians (88.3%) chose synthetic material for the MUS, while only a small minority preferred autologous fascia (6.4%) and biologic materials (5.3%). As for the urethral bulking agents, the top three choices were polyacrylamide (Bulkamid) (26.6%), polytetrafluoroethylene (Teflon, PTFE) (23.4%), and bovine collagen (Contigen) (22.3%).

## 4. Discussion

The present study showed that the Jordanian OBGYNs preferred that the first-line treatment options follow the National Institute for Health and Care Excellence (NICE) 2023 guidelines [10], with 43.6% favoring lifestyle/behavioral therapy and 40.4% preferring pelvic floor muscle physiotherapy [9]. This preference is corroborated through research demonstrating improved quality of life and decreased urinary leakage measured by a 1 h pad test, Incontinence Quality of Life (IQOL), quality of life questionnaires, and King’s health questionnaire using pelvic floor physiotherapy [17]. Furthermore, 14.9% of our target population preferred an initial surgical procedure instead of conservative measures. The latter preference is also evidenced by a multicenter, randomized study on 230 women with moderate to severe SUI whereby the initial MUS resulted in higher rates of subjective improvement and subjective and objective cure rates [18]. 

The Jordanian OBGYNs demonstrated that the MUS-TO is the method of choice for treating SUI at 46.8%, followed by the MUS-RP and Kelly plication at 17%. Our findings reinforce the research showing that the healthcare professionals in a tertiary-level hospital in the UK opted for the MUS as their preferred surgical approach for SUI patients [19]. On the other hand, a recent updated guideline study reported that there were conflicting studies on which one was more favorable between the MUS-TO and MUS-RP [9]. For instance, a slight advantage toward the MUS-RP was observed in a 5-year follow-up [20], whereas, in a systemic review study, no difference was found between the two treatments regarding patient satisfaction, QOL, and objective and subjective cure [21]. However, it is also essential to note that it is not only the preference of medical professionals but also patients demonstrated that a surgical curative approach was preferred when faced with severe SUI [22]. It was also observed that, when the patients were provided a scenario with a less invasive approach with similar curative rates, they opted for the less invasive option.

The less invasive advancements within the field include the use of urethral bulking agents (UBAs) [23]. Our data showed that 72.4% of the respondents have never used UBAs for SUI, and 27.6% opted for UBAs either initially or following the initial procedure failure. A systematic review and meta-analysis including 710 patients comparing the safety and effectiveness of UBAs and surgical methods demonstrated that the UBAs were less effective in terms of subjective improvement, and there was no significant difference in terms of complications following the intervention [24], thus demonstrating that the medical professional preference for surgery is validated through the merits of the procedure in terms of the balance between the outcome and complications.

Our findings showed that the diagnostic modality preference was associated with the fellowship experience. Furthermore, the primary selection of MUS-TO surgery and no need for SUI surgery when POP is present and with no symptoms of SUI were linked with or without fellowship/subspecialty training. Finally, the female OBGYNs were in favor of not performing SUI surgery when POP and SUI with no symptoms were present. This gender difference may be psychological. However, other socioeconomic factors did not affect the respondent preference when analyzing the practice settings. It is also worth noting that there was no difference in preference with respect to the geographic location or age of the respondent. It is also vital to demonstrate that our data are comparable to those in high-income countries (HICs), further demonstrating the lack of socioeconomic variables affecting the preference rather than substantiating the preference in terms of the merit of the procedure alone [19].

The European Association of Urology, NICE, and Cochrane guidelines posit that the initial surgical intervention offered to women with SUI should be an MUS [25]. Furthermore, the American Urological Association and Society of Urodynamics, Female Pelvic Medicine & Urogenital Reconstruction guidelines recommend that, if an MUS procedure is to be carried out, the surgeons should opt for a transobturator or retropubic approach [9,26]. Therefore, it is essential to substantiate the respondent’s preference for the use of the MUS. However, post-operative mesh-related complications such as erosion into adjacent structures, pain during intercourse, and chronic pain were observed [19]. Although the US Food and Drug Administration and National Health Services England have banned the use of transvaginal mesh including the MUS, the incidence of these complications/symptoms to occur is low and the MUS is still the preferred surgical option for the healthcare professionals in the United Kingdom tertiary hospital [19]. Furthermore, several systematic review studies confirmed that the MUS-RP or MUS-TO are the preferred treatments for SUI [27,28]. 

It is vital to acknowledge the limitations of this study. Although we addressed all the OBGYNs in Jordan, most of our respondents were from one geographical region, Amman. Thus, one cannot generalize these data with certainty to the remainder of the country and other LMICs, especially when SUI is prevalent in the rural areas of Jordan and the incomes of the families are much less than in the capital, Amman [14,15]. More research would be essential comparing the medical professional preference in terms of SUI management between the LMICs and HICs. Secondly, the target population comprised OBGYNs only, posing another limitation; one could argue that the spread of preference might differ within other specialties, as demonstrated through other papers [19]. Therefore, it would be ideal if research was carried out using OBGYNs, urogynecologists, urologists, and actual patients to measure whether the medical specialty or lack thereof affects the preference. Interestingly, it is worth noting that the data gathered with regard to the management of SUI show a preference for treatment options that offer the highest symptomatic relief as well as ones with lower complication rates, which is in line with an MUS procedure [27,28,29,30].

## 5. Conclusions

The Jordanian OBGYNs preferred using lifestyle/behavioral therapy and pelvic floor muscle physiotherapy as the first-line treatment to manage SUI, which is according to the NICE guidelines. The MUS, mainly the MUS-TO, would be the most frequently preferred surgical choice. Furthermore, since 70% of OBGYNs refer recurrent SUI post-surgery to urogynecologists, this study supports the need for more specialization in urogynecology. Therefore, more efforts from Jordan’s and LMICs’ ministries of health are essential to acquire subspecialty training to tackle SUI, which is affecting millions of women’s quality of life.

## Figures and Tables

**Figure 1 healthcare-12-01489-f001:**
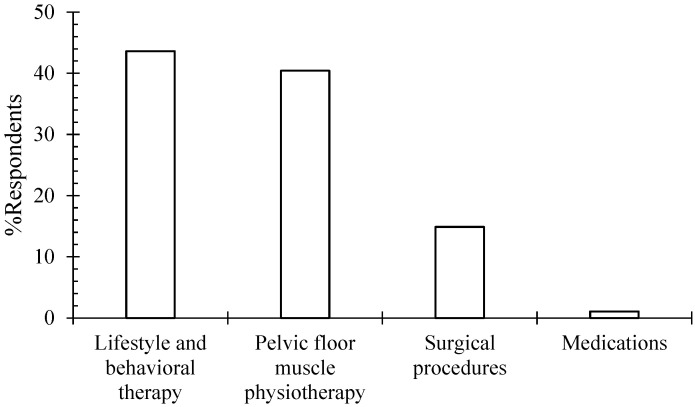
Breakdown of Jordanian OBJYNs’ initial SUI management preferences.

**Figure 2 healthcare-12-01489-f002:**
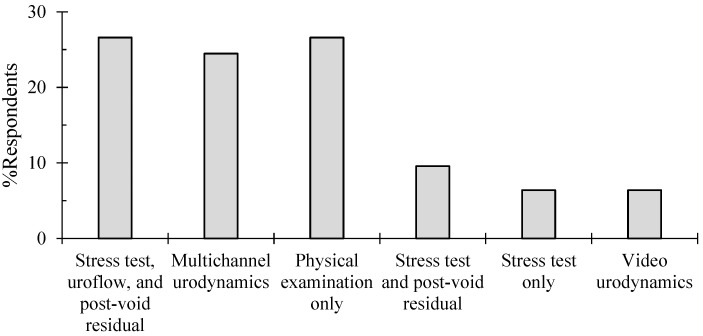
Breakdown of Jordanian OBJYNs’ pre-surgical diagnostic modality preferences of SUI.

**Figure 3 healthcare-12-01489-f003:**
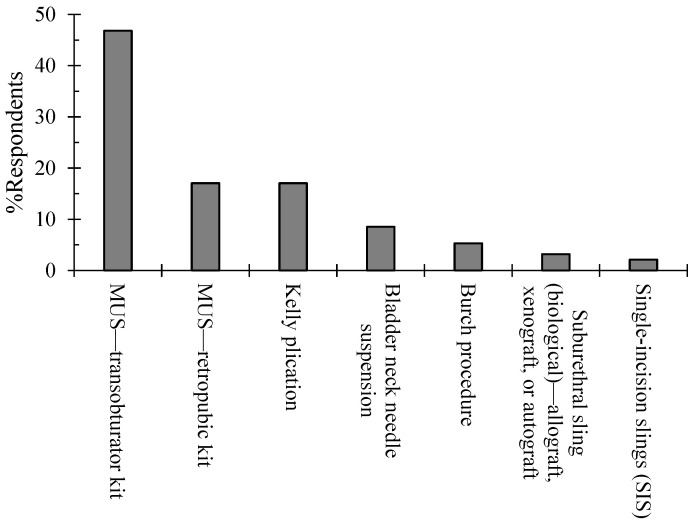
Jordanian OBJYNs’ primary surgical treatment options for SUI.

**Table 1 healthcare-12-01489-t001:** The demography, practice, and experience of the OBGYN respondents.

Parameters	Characteristics	Number	Percentage (%)
Age	<30 years old	7	7.4
	30–40 years old	28	29.8
	41–55 years old	34	36.2
	>55 years old	25	26.6
Gender	Female	60	63.8
	Male	34	36.2
Specialty	Gynaecology	86	91.5
	Urogynecology	8	8.5
Practice setting	Private Practice	67	71.3
	Ministry of Health	10	10.6
	Academic	10	10.6
	Other	7	7.4
Location of Practice	Amman	83	88.3
	Alzarqa	4	4.3
	Irbid	3	3.2
	Albalqa	2	2.1
	Madaba	1	1.1
	Alkarak	1	1.1
Years Out of Fellowship/Subspecialty Training	<5 years	9	9.6
	5–10 years	9	9.6
	>10 years	21	22.3
	No fellowship/subspecialty training	55	58.5

## Data Availability

Data is contained within the article or Appendix A.

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
