# Peer review of "Management of Stress Urinary Incontinence by Obstetricians and Gynecologists in Jordan: A Nationwide Survey Study"

_healthcare, 2024, doi:10.3390/healthcare12151489_

Round 1
Reviewer 1 Report
Comments and Suggestions for Authors
Qatawneh et al. report a nationwide survey among Jordanian obstetricians and gynecologists (OBGYNs) to investigate the management of stress urinary incontinence. While the topic is significant, several issues need addressing before it is deemed worthy of publication.
a. The title is clear and concise. However, adding the study population can provide additional context.
b. In the abstract, “OBGYNs” should be spelt out as “Obstetricians and Gynaecologists” the first time it is mentioned, followed by the abbreviation (OBGYNs) in parentheses.
c. The authors may want to rephrase Lines 15-17 for clarity.
d. In Line 21, the authors should consider removing the “tended to” and just use the past tense form of “prefer”.
e. The study’s adherence to ethical standards, including obtaining IRB approval and ensuring participant confidentiality and anonymity, is commendable. However, it would be helpful to include more details on how these ethical considerations were implemented in practice.
f. The survey questionnaire, being the primary tool of data collection, was not validated. Without validation, the reliability and validity of the responses are uncertain.
g. Distributing the survey via WhatsApp may have introduced selection bias, as not all OBGYNs may use WhatsApp or check it regularly for professional purposes. Alternative distribution methods, such as email or in-person distribution, could improve accessibility and response rates.
h. The response rate of 39.2% may not be representative of the entire population of OBGYNs in Jordan. A higher response rate would increase the generalizability of the findings.
i. The majority of the respondents (88.3%) were from the capital, Amman, which could introduce a location bias.
j. The “do” in Line 95 could be changed to “perform”.
k. The word “have” in Line 99 should be changed to its past tense form.
l. The word “However” is used twice in close succession (Lines 116-118). The authors may want to consider revising to avoid repetition and improve flow.
m. The transition from paragraph to paragraph (Lines 133 to 162) could be smoother. Using transitional phrases could improve the flow.
Comments on the Quality of English LanguageThe authors may want to make some adjustments to their grammatical presentation.
Author Response
Qatawneh et al. report a nationwide survey among Jordanian obstetricians and gynecologists (OBGYNs) to investigate the management of stress urinary incontinence. While the topic is significant, several issues need addressing before it is deemed worthy of publication.
- The title is clear and concise. However, adding the study population can provide additional context.
Added.
- In the abstract, “OBGYNs” should be spelt out as “Obstetricians and Gynaecologists” the first time it is mentioned, followed by the abbreviation (OBGYNs) in parentheses.
It was spelt out line 12 in the original version.
- The authors may want to rephrase Lines 15-17 for clarity.
The statement was rephrased.
- In Line 21, the authors should consider removing the “tended to” and just use the past tense form of “prefer”.
Rephrased.
- The study’s adherence to ethical standards, including obtaining IRB approval and ensuring participant confidentiality and anonymity, is commendable. However, it would be helpful to include more details on how these ethical considerations were implemented in practice.
We applied what has been stated in section 2.2 that “the data provided will only be used for research purposes and will be dealt with confidentially. None of the questions address your identity, which will be anonymous throughout. Proceeding in completing this questionnaire will be taken as consent to participate in this study, and you are always free to withdraw at any point.”
- The survey questionnaire, being the primary tool of data collection, was not validated. Without validation, the reliability and validity of the responses are uncertain.
- Distributing the survey via WhatsApp may have introduced selection bias, as not all OBGYNs may use WhatsApp or check it regularly for professional purposes. Alternative distribution methods, such as email or in-person distribution, could improve accessibility and response rates.
It all depends on the culture of the country. In Jordan, the first communication tool is WhatsApp. Every person uses it including, physicians, diagnostic laboratories, Hospitals, and Jordanian government and ministries.
- The response rate of 39.2% may not be representative of the entire population of OBGYNs in Jordan. A higher response rate would increase the generalizability of the findings.
We agree with the reviewer comment, but we have repeatedly (3 times) encouraged OBJYNs to respond to the survey.
- The majority of the respondents (88.3%) were from the capital, Amman, which could introduce a location bias.
We agree with the reviewer and this was one the limitation of the study (line 223).
- The “do” in Line 95 could be changed to “perform”.
Changed.
- The word “have” in Line 99 should be changed to its past tense form.
Changed.
- The word “However” is used twice in close succession (Lines 116-118). The authors may want to consider revising to avoid repetition and improve flow.
The two sentences were rephrased.
- The transition from paragraph to paragraph (Lines 133 to 162) could be smoother. Using transitional phrases could improve the flow.
The beginning of the two middle paragraphs were modified to improve the flow.

Reviewer 2 Report
Comments and Suggestions for Authors
Thank you for this important paper, Management of Stress Urinary Incontinence in Jordan: A Nationwide Survey Study. It is well-written and provides important information. However: In the discussion, please report how your findings of current practice in your country compare to similar studies in other countries or continents.
Author Response
Thank you for this important paper, Management of Stress Urinary Incontinence in Jordan: A Nationwide Survey Study. It is well-written and provides important information. However: In the discussion, please report how your findings of current practice in your country compare to similar studies in other countries or continents.
We thank the reviewer for the above comments. Although there are many research studies on SUI, there are very few studies on OBGYN’s management of SUI that had a similar approach as presented in our manuscript. Therefore the comparison would not be possible. However, we would like to point out that we referred to studies whereby OBGYNs practice (lifestyle/behavioral therapy and pelvic floor muscle physiotherapy or type of surgery) in Jordan is similar to NICE guidelines, and other studies performed in other countries such as (References 9, 10, 18, 19, 25, 27 and 28).

Reviewer 3 Report
Comments and Suggestions for Authors
- The article addresses the treatment options employed by practitioners through a nationwide survey in a low-middle-income country. Stress urinary incontinence is common in women with a significant impact on quality of life. Hence, uniform practice, proper use of nonsurgical options, and referrals would be essential in LMICs.
- The abstract is structured, well-written, and representative of the study. However, it is suggested to include the headings.
- The title is appropriate and the introduction describes the importance of the research.
- Methodology is elaborate, well-described, and clear including participant selection, and details of data analysis. However, the source/ validation of the questionnaire is not mentioned.
- The results are well described. The table and charts were appropriately used.
· The discussion is appropriate and explores the importance of the findings, by comparing it to the existing literature.
- The conclusion is consistent with the results and interpretation.
- The references are appropriate, and well-cited. However, it may need to be modified to MDPI style.
- Minor modifications required in language and grammar
- E.g. paragraph-1- Such high affected people indicating high incidence and prevalence.
--Paragraph-2- Assessment of patients begins with a focused history and physical exam and can include post-residual urine volume, urine testing, and urodynamic testing.
Line-184- …patients demonstrated that a surgical curative approach when faced with severe SUI
- Lower and Middle Income countries could be LMIC in place of LIMC
Comments on the Quality of English Language
Quality is good. Minor modifications are suggested.
- Minor modifications required in language and grammar
- E.g. paragraph-1- Such high affected people indicating high incidence and prevalence.
--Paragraph-2- Assessment of patients begins with a focused history and physical exam and can include post-residual urine volume, urine testing, and urodynamic testing.
Line-184- …patients demonstrated that a surgical curative approach when faced with severe SUI
- Lower and Middle Income countries could be LMIC in place of LIMC
Author Response
- The article addresses the treatment options employed by practitioners through a nationwide survey in a low-middle-income country. Stress urinary incontinence is common in women with a significant impact on quality of life. Hence, uniform practice, proper use of nonsurgical options, and referrals would be essential in LMICs.
- The abstract is structured, well-written, and representative of the study. However, it is suggested to include the headings.
Added.
- The title is appropriate and the introduction describes the importance of the research.
- Methodology is elaborate, well-described, and clear including participant selection, and details of data analysis. However, the source/ validation of the questionnaire is not mentioned.
In the revised version, we performed a validity of the questionnaire using the 2-tailed Pearson correlation (line 92).
- The results are well described. The table and charts were appropriately used.
- The discussion is appropriate and explores the importance of the findings, by comparing it to the existing literature.
- The conclusion is consistent with the results and interpretation.
- The references are appropriate, and well-cited. However, it may need to be modified to MDPI style.
Corrected.
- Minor modifications required in language and grammar
- E.g. paragraph-1- Such high affected people indicating high incidence and prevalence.
The statement is modified to “Such highly affected people indicate high incidence and prevalence.”
--Paragraph-2- Assessment of patients begins with a focused history and physical exam and can include post-residual urine volume, urine testing, and urodynamic testing.
The statement is modified to “The patient’s assessment begins with a focused history and physical exam and can include different investigations, such as post-residual urine volume, urine testing, and urodynamic testing”.
Line-184- …patients demonstrated that a surgical curative approach when faced with severe SUI
The statement is modified to “However, it is also essential to note that it is not only the preference of medical professionals, but also patients demonstrated that a surgical curative approach when faced with severe SUI”.
- Lower and Middle Income countries could be LMIC in place of LIMC
Replaced!

Reviewer 4 Report
Comments and Suggestions for Authors
Thanks for the opportunity to review this manuscript. The authors endeavor to identify and describe the current management strategies employed by OBGYNs in Jordan through a cross-sectional survey-based study. The study population is well defined, as are the descriptive goals and rationale of the study.
The authors need to address a few issues:
-response rate for a survey is usually considered # responses / # of invitations. Clarify how you calculated response rate and what you mean by 497 "could be reached"- did they respond and indicate whether they did surgery or not? Then, of those 240 from 497 who responded to you, 94 completed the survey?
-conclusions must follow from results. Unless I am missing something, you have identified that the majority of respondents are meeting international standards of care RE: offering conservative therapies first, then surgical therapies, and when they offer surgical therapies, they are offering the correct/most appropriate ones on average? If this is the case, then your conclusions should maybe be more positive; you have presented evidence that, despite being situated in an LMIC environment, your providers are offering the international standard of care. If you DON'T feel that this was the case, and that your conclusion is that you need more resources, more training etc, then you need to make this explicitly clear by demonstrating a mismatch between a standard of care and your respondents' care.
-in methods, you indicate that you will be presenting descriptive statistics like frequencies and percentages etc and that "chi squared analyses were conducted on continuous variables" but then you mention multiple times about there being no 'associations' between multiple different variables. Did you do regression analysis testing for associations between specific variables, accounting for different confounders? that would be the best way to look at specific differences you might be interested in. Or, did you just do a lot of pairwise comparisons between different variables? Either way, this element of your manuscript needs clarification. There should also be at least some description of your planned analysis in the methods (i.e. were you going into this wondering if subspecialty fellowship training in urogynecology vs. generalist OBGYN training was associated with differences in surgical procedure offered, accounting for surgeon age and geographic location? If so, again this kind of question would be best addressed with regression analysis.
Overall, this is generally well presented, but please address the above issues to elevate the quality and impact of the manuscript. See attached marked up pdf as well

Author Response
Thanks for the opportunity to review this manuscript. The authors endeavor to identify and describe the current management strategies employed by OBGYNs in Jordan through a cross-sectional survey-based study. The study population is well defined, as are the descriptive goals and rationale of the study.
The authors need to address a few issues:
-response rate for a survey is usually considered # responses / # of invitations. Clarify how you calculated response rate and what you mean by 497 "could be reached"- did they respond and indicate whether they did surgery or not? Then, of those 240 from 497 who responded to you, 94 completed the survey?
From the 497 OBGYNs, 240 performed the gynecological surgeries, and therefore, 257 were not eligible for the study. So, 94 completed the survey out of 240 eligible for the survey (39.2%). In the revised version, we changed the sentence to “Out of 804 registered Jordanian OBGYNs, 497 could be reached, and 240 performed gynecological surgeries and therefore were eligible for the study. Out of the 240 OBGYNs, 94 completed the survey, giving a response rate of 39.2%.”
-conclusions must follow from results. Unless I am missing something, you have identified that the majority of respondents are meeting international standards of care RE: offering conservative therapies first, then surgical therapies, and when they offer surgical therapies, they are offering the correct/most appropriate ones on average? If this is the case, then your conclusions should maybe be more positive; you have presented evidence that, despite being situated in an LMIC environment, your providers are offering the international standard of care. If you DON'T feel that this was the case, and that your conclusion is that you need more resources, more training etc, then you need to make this explicitly clear by demonstrating a mismatch between a standard of care and your respondents' care.
Our study showed that the Jordanian OBGYN preference was using lifestyle/behavioral therapy and pelvic floor muscle physiotherapy as the first line to manage SUI, which is according to NICE guidelines. Furthermore, MUS, and mainly MUS-TO, would be the most frequent preferred surgical choice. However, with a high percentage would refer recurrent SUIs to a specialist suggest that more training is required. Thus, we have included both in the conclusion which is similar to the reviewer’s points above.
-in methods, you indicate that you will be presenting descriptive statistics like frequencies and percentages etc and that "chi squared analyses were conducted on continuous variables" but then you mention multiple times about there being no 'associations' between multiple different variables. Did you do regression analysis testing for associations between specific variables, accounting for different confounders? that would be the best way to look at specific differences you might be interested in. Or, did you just do a lot of pairwise comparisons between different variables? Either way, this element of your manuscript needs clarification. There should also be at least some description of your planned analysis in the methods (i.e. were you going into this wondering if subspecialty fellowship training in urogynecology vs. generalist OBGYN training was associated with differences in surgical procedure offered, accounting for surgeon age and geographic location? If so, again this kind of question would be best addressed with regression analysis.
As suggested by the reviewer, we performed a univariate analysis between each question (dependent variable) and the fixed parameter. We added this statement in the Statistical Analysis section, and we added paragraphs to describe the results for those that should significance association (lines 131-135 and 139-143)).
Overall, this is generally well presented, but please address the above issues to elevate the quality and impact of the manuscript. See attached marked up pdf as well.
We thank the reviewer for the comments that were addressed to elevate the quality and impact of the manuscript.

Round 2
Reviewer 4 Report
Comments and Suggestions for Authors
Thanks to the authors and editors for the opportunity to review the revisions following peer review.
Thank you for addressing the concerns I outlined.
You have adequately clarified the survey response numbers, the statistical analysis, and conclusions. Specifically, I feel that the slight wording change of specifying that this study provides support that there may be a need for more urogynecolgists if 70% of OBGYNs are referring recurrent SUI post surgery is a bit clearer.
That said, how does the reader (or the authors) know that there are not enough urogynecologists in Jordan already? And what is the appropriate percentage of cases that should be referred to a urogynecologist post-operatively with recurrent SUI? Locally in North America, we would refer 100% of those cases to urogynecology. I was actually surprised to see that 30% of general OBGYNs would not refer to a subspecialist if they had a failed surgery for SUI. Or, are they not referring because they cant refer because they don't have local access to urogynecologists?
Comments on the Quality of English LanguageMinor grammatical issues that do not detract much from the quality of the manuscript
Author Response
Responses to Reviewer 4 Comments:
You have adequately clarified the survey response numbers, the statistical analysis, and conclusions. Specifically, I feel that the slight wording change of specifying that this study provides support that there may be a need for more urogynecologists if 70% of OBGYNs are referring recurrent SUI post-surgery is a bit clearer.
We changed the sentence in the conclusion (line 254 to 258) to “Furthermore, since 70% of OBGYNs refer recurrent SUI post-surgery to urogynecologists, this study supports the need for more specialization in urogynecology. There-fore, more efforts from Jordan’s and LMIC’s Ministries of Health are essential to acquire subspecialty training to tackle SUI, which is affecting millions of women’s quality of life.”
That said, how does the reader (or the authors) know that there are not enough urogynecologists in Jordan already?
Table 1 shows that only 8.5% have urogynecology as a specialty. Furthermore, in Jordan, we do not have a subspeciality urogynecology training program.
And what is the appropriate percentage of cases that should be referred to a urogynecologist post-operatively with recurrent SUI?
All patients who had recurrent incontinence should be referred.
Locally in North America, we would refer 100% of those cases to urogynecology. I was actually surprised to see that 30% of general OBGYNs would not refer to a subspecialist if they had a failed surgery for SUI. Or, are they not referring because they cant refer because they don't have local access to urogynecologists?
We do not know why general gynecologists don’t refer recurrent cases if the SUI surgery failed, and the present study did not address such information.
